# Development and Validation of a Knowledge, Attitude and Practice Questionnaire on Antibiotic Use in Arabic and French Languages in Lebanon

**DOI:** 10.3390/ijerph19020687

**Published:** 2022-01-08

**Authors:** Narmeen Mallah, Rubén Rodríguez-Cano, Danielle A. Badro, Adolfo Figueiras, Francisco-Javier Gonzalez-Barcala, Bahi Takkouche

**Affiliations:** 1Department of Preventive Medicine, University of Santiago de Compostela, 15705 Santiago de Compostela, Spain; narmeen.mallah@usc.es (N.M.); adolfo.figueiras@usc.es (A.F.); bahi.takkouche@usc.es (B.T.); 2WHO Collaborating Centre for Vaccine Safety, 15706 Santiago de Compostela, Spain; 3Genetics, Vaccines and Pediatric Infectious Diseases Research Group (GENVIP), Instituto de Investigación Sanitaria de Santiago de Compostela, 15706 Santiago de Compostela, Spain; 4Centro de Investigación Biomédica en Red de Enfermedades Respiratorias (CIBER-ES), 28029 Madrid, Spain; 5PROMENTA Research Center, Department of Psychology, University of Oslo, 0315 Oslo, Norway; r.r.cano@psykologi.uio.no; 6Faculty of Health Sciences, American University of Science and Technology, Beirut 1100, Lebanon; dbadro@aust.edu.lb; 7INSPECT-LB, Institut National de Santé Publique, Epidémiologie Clinique et Toxicologie, Beirut 1100, Lebanon; 8Centro de Investigación Biomédica en Red de Epidemiología y Salud Pública (CIBER-ESP), 28029 Madrid, Spain; 9Health Research Institute of Santiago de Compostela (IDIS), 15706 Santiago de Compostela, Spain; 10Department of Medicine, University of Santiago de Compostela, 15705 Santiago de Compostela, Spain; 11Department of Respiratory Medicine, University Hospital of Santiago de Compostela (CHUS), 15706 Santiago de Compostela, Spain

**Keywords:** antibiotics, Arabic, French, knowledge–attitude–practice, Lebanon, questionnaire, validation

## Abstract

Objectives: Validated knowledge–attitude–practice (KAP) questionnaires are essential to design and evaluate intervention programs on antibiotic use. Recently, we validated the first KAP questionnaire on antibiotics in Spain. Cross-cultural adaptation and validation of research tools increase their universal usefulness. Here, we aimed to validate the questionnaire in a developing country with different socioeconomic characteristics from that of Spain. Methods: We translated the previously developed KAP-questionnaire into Arabic and French, tailored it and then validated it in adult population in Lebanon. The item content validity index (I-CVI), scale content validity index (S-CVI/Ave) and modified Kappa (k*) were calculated. The construct validity of the questionnaire was evaluated using confirmatory factorial analysis (CFA, N = 1460) and its reliability was assessed using intraclass correlation coefficients (ICC, N = 100) and Cronbach’s alpha statistic. Results: ICV-I (>0.78), k* (equal to ICV-I for all items) and S-CVI/Ave (≥0.95) confirmed the questionnaire content validity. Pilot testing (N = 40) and face validity showed the understandability of the questionnaire by the population. Test–retest reliability analysis (N = 100) yielded ICC ≥ 0.59 for all knowledge and attitude items, showing the capacity of the questionnaire to generate reproducible results. CFA evidenced adequate fit of the chosen model, thus establishing the construct validity of the questionnaire (root mean squared error approximation = 0.053, standardized root mean square residual = 0.045, comparative fit index = 0.92 and Tucker–Lewis index = 0.90). The questionnaire showed an acceptable internal reliability (Cronbach’s alpha = 0.62) and was highly accepted in Lebanon (response rate = 96% and item response rates ≥ 94%). Conclusions: The validity of the KAP-questionnaire on antibiotics in Arabic and French was demonstrated in Lebanon.

## 1. Introduction

Knowledge–attitudes–and–practices (KAP) modelled questionnaires are extensively used in studies about the use of antibiotics [1,2,3]. Using these instruments, researchers uncover erroneous knowledge of the population on antibiotics, assess attitudes towards those drugs and evaluate the adequateness of practices of antibiotic use [4]. Antibiotics are misused when taken, purchased or shared without medical prescription or when prescribed by the doctor, but the patient fails to adhere to the treatment guidelines on timing, dosage and duration [5,6]. 

Antibiotic resistance is a multi-faceted international public health issue that has been exacerbated by antibiotic misuse. It has challenged modern therapies and imposed devastating economic loses [7]. Approximately seven hundred thousand individuals die each year due to infections with antibiotic-resistant bacteria and the attributed mortality rate is projected to reach 10 million by 2050, exceeding that of major diseases including cancer [8].

The use of KAP questionnaires on antibiotics is not only useful at baseline to examine these determinants in a population at a specific moment, but they also represent the backbone for assessing the need, and designing and evaluating educational intervention programs to enhance the rationale use of antibiotics [9]. It is essential, therefore, to base these evaluations on validated tools that generate reliable results.

We have recently reported on the development and validation of the first KAP questionnaire on the use of antibiotics by the general adult population in Spain [10]. Questionnaires that had been validated in a specific country require prior testing and adaptation before they may be used in another country with different language and culture [11,12]. 

In the present study, we aimed to assess the validity of the KAP questionnaire in Lebanon, after adapted and translated into Arabic and French from the questionnaire previously developed and validated in Spain. Both countries differ widely in language, culture, public health system and socio-demographic characteristics. Arabic and French are widely spoken languages in the world. Arabic is spoken by 420 million individuals worldwide and is the official or co-official language in 25 countries, making it the fifth most spoken language worldwide [13]. French is spoken in 57 member states of the Francophonie [14], and also has an important position in several Arab countries including Algeria, Lebanon, Morocco and Tunisia [15]. 

## 2. Methods

### 2.1. Study Setting and Population

The previously validated KAP questionnaire on antibiotic use by the general population [10], was translated forward and backward into Arabic and French by native multilingual researchers (see Appendix A). The translated questionnaire was then fully validated in 2019 in Lebanon in a population of adults. In Lebanon, Arabic is the native language, and French is totally or partially spoken by half of the population and is considered the second language in the country [15]. French is also the first teaching language in 70% of primary schools in Lebanon [15]. The English version of the questionnaire is published elsewhere [10].

The validation study was carried out in a population of adults, consisting of parents of children attending schools in Beirut, Lebanon. This population was chosen on the basis that schools provide convenient access to an administratively defined adult population in Lebanon. 

Lack of collaboration in research studies has been widely observed in Lebanon, where most of previous studies had relied on convenience samples with small sizes. Therefore, to ensure a sufficient population size, more than 200 private and public schools in Beirut were contacted via phone, email and/or on-site visit to seek their collaboration in the study. Schools were reluctant to cooperate mainly because of internal administrative difficulties such as lack of personnel and overlapping between study period and prescheduled school activities including exams. Some schools requested incentives for participation which was not compatible with the ethical procedure of the study. Finally, eleven schools joined the study.

The researchers did not have access to school registries; thus, the schools decided the number of questionnaires to be distributed based on their administrative capacities and the number of unique parents (one questionnaire per family). A total of 1460 questionnaires were distributed.

Before questionnaire circulation, parents were informed about the study objective and on the scheduled questionnaire delivery date. The parents were then provided with a printed copy of the questionnaire in the language chosen by the school (Arabic or French). They were instructed that only one of the parents (either the mother or the father) could complete the questionnaire. In the first working day of the week, the questionnaires were given to the parents in a sealed envelope by their eldest child who was attending the school. The same day, the parents were notified about the questionnaire delivery. Questionnaires were then sent back to the school in a closed envelope, in person or by their child. The questionnaires were collected until the last working day of the same week by the school personnel in order to avoid questionnaire loss during the week-end. Subsequently, two researchers (NM and DAB) gathered the questionnaires from the schools.

### 2.2. Validation Procedure

The validation procedure of the original questionnaire in Spanish is described in detail elsewhere [10]. The questionnaire involves 17 *knowledge* and *attitude* statements that are answered using a 0–10 Likert scale. The numbers zero and 10 represent the lowest and highest levels of agreement, respectively. The questionnaire also encompasses *practice* and sociodemographic characteristics questions that are completed by selecting one or more of a given possible answers. The sociodemographic characteristics block of the original questionnaire was tailored to best fit the Lebanese population, where new questions regarding income, spouse employment, spouse education level, and access to healthcare were added [10].

### 2.3. Content Validity

Each of the translated Arabic and French versions of the questionnaire was separately examined by a panel of nine experts for its adequateness and completeness [16,17]. The experts were Lebanese bilingual (Arabic and French) with a specialty in pharmacy, medicine, or miscellaneous health sciences. The experts rated the clarity and relevancy of each item of the questionnaire on a scale of one (lowest) to four (highest). The content validity of the questionnaire was then determined by calculating the item content validity index (I-CVI), scale content validity index average (S-CVI/Ave) and modified kappa (k*) statistic [18,19].

### 2.4. Face Validity

Two research members (NM and BT) reviewed the questionnaire for its clarity and completeness. The face validity of a questionnaire is established when it seems to measure what it is intended to measure [17], i.e., knowledge, attitudes and practices of antibiotic use by the general adult population.

### 2.5. Pilot Testing

Each of the Arabic and French versions of the questionnaire was pilot-tested in a sample of 20 adults who were recruited from different non-health-related professions and who belonged to different socioeconomic levels. Participants were asked to comment on the clarity, format, and length of the questionnaire and to suggest modifications for questionnaire improvement.

### 2.6. Reliability

Knowledge and attitudes are stable characteristics over short time periods. Therefore, their reliability was tested by distributing the questionnaire twice on 100 parents of schoolchildren who accepted to answer the questionnaire on two occasions within one month. The parents were provided with the version of the questionnaire selected by the school (Arabic or French). The intraclass correlation coefficient (ICC) relative to the average measure of the two-way mixed-effects model was then calculated for each of the 17 Knowledge and Attitude items [20]. An item was retained in the questionnaire if its ICC value was >0.4 [21].

### 2.7. Construct Validity

Confirmatory factorial analysis (CFA) was carried out to examine the validity of the knowledge and attitude construct of the questionnaire. The questionnaire was distributed to 1460 parents of children registered in 11 schools in Beirut, Lebanon. Knowledge and attitude items were allocated to their respective factors following the pattern of the original questionnaire validated in Spain [10]. They were distributed into three factors: (1) *knowledge*, (2) *personal attitudes towards antibiotics* and (3) *attitudes towards healthcare providers*. We standardized the factors by constraining them to a mean of 0 and a variance of 1, examined the standardized residual correlations between items and applied the modification indexes method to better allocate items to each factor, and, consequently, to enhance the fit of the model [22,23]. We applied the full information maximum likelihood method to handle the missing data. The goodness of fit of the model was evaluated by calculating the following statistics: root mean squared error approximation (RSMEA, acceptable if <0.08), comparative fit index (CFI, acceptable if ≥0.90), Tucker–Lewis index (TLI, acceptable if ≥0.90) and standardized root mean square residual (SRMR, acceptable if <0.08) [24]. We also calculated the Chi-squared (*X*^2^) statistic value as well as Akaike information criterion (AIC), Bayesian information criterion (BIC) and sample-size adjusted BIC (aBIC). These indicators reveal the relative amount of information lost by a model. The lower *X*^2^, AIC, BIC and aBIC values the better the quality of the model [25]. 

### 2.8. Questionnaire Overall Reliability, Acceptability and Item Response Rate

Using the data collected from the 1460 parents, the overall reliability of the questionnaire was tested by calculating Cronbach’s alpha index (acceptable if >0.6) [26,27]. The acceptability of the questionnaire was also determined by calculating the response rate, i.e., the percentage of answered questionnaires. Finally, the acceptability of the items of the questionnaire by the Lebanese general adult population was inspected by calculating the proportion of answered questions [28,29,30,31].

## 3. Results

### 3.1. Content Validity

I-CVI values ranged between 0.78 and 1.00, indicating that the panel of nine experts found that the items of the questionnaire translated into Arabic and French are clear, easily understandable, and related to KAP on antibiotic use. All items of the two versions of the questionnaire (Arabic and French) showed K* statistic > 0.75 and equal to I-CVI, revealing the unlikeliness of agreement by chance between the nine experts on the clarity and the relevancy of the items of the questionnaire. S-CVI/Ave was≥ 0.95, thus demonstrating the content validity of the scale.

### 3.2. Face Validity and Pilot Testing

The researchers found that the items of the questionnaire measured the target topic (KAP on antibiotic use), establishing therefore the questionnaire face validity. The 40 participants of the pilot testing answered the questionnaire in its totality (20 in Arabic and 20 in French). The participants did not suggest any questionnaire modification and reported an overall satisfaction about the ease of answering and the comprehension of the wording of the questions in Arabic as well as in French.

### 3.3. Reliability

Ninety-one adults answered the Arabic and French questionnaires on two occasions. ICC values were≥ 0.59 for all knowledge and attitudes items, indicating their capacity to generate reproducible results (Table 1).

### 3.4. Construct Validity

Various models of knowledge and attitude items were examined according to theoretical and logical backgrounds as well as indications of the method of modification indices.

Model 1. The 17 knowledge and attitude items were assigned to three factors (*knowledge*, *personal attitudes towards antibiotics*, and *patient attitudes towards healthcare provider*) following the structure of the model previously validated in the Spanish population. Items Q1, Q2, Q4, Q6–8 and Q11 were attributed to *knowledge*, items Q3, Q5, Q9, Q12 and Q14 were assigned to *personal attitudes towards antibiotics*, and items Q10, Q13, Q15, Q16, and Q17 were allocated to *patient attitudes towards healthcare provider*. In this model, item Q7 *“**each infection needs a different antibiotic”* did not significantly load on the *knowledge* factor (*p*-value = 0.510), and the model indicators of goodness of fit did not show an adequate fit (Table 2).

Model 2. This model followed the same structure as Model 1, but item Q7 was removed from the *knowledge* factor. Q17 “*When you buy antibiotics, the pharmacist tells you about the importance of correct therapeutic compliance/adherence*” was also attributed to *Knowledge* in addition to *patient attitudes towards healthcare provider* factors. All items significantly loaded on their respective factors (*p*-value < 0.005) and the model goodness of fit indicators showed improvement, but they were still not acceptable (Table 2).

Model 3. In this model, item Q10 “*I take the antibiotics according to the doctor’s instructions*” was moved to *personal attitudes towards antibiotics* factor, and item residuals of Q10 and Q13, Q15 and Q16 which belong to *patient attitudes towards healthcare provider factor* were correlated. The goodness of fit indicators had improved but the model could still be enhanced further (Table 2).

Model 4. At this stage, item Q10 was attributed to the factor *personal attitudes towards antibiotics* in addition to the factor *patient attitudes towards healthcare provider,* and the residuals of items Q9, Q10 and Q11 were also correlated. All items showed a significant load on their corresponding factors, and most of the goodness of fit indicators were acceptable (Table 2 and Table 3).

Model 4 showed an acceptable fit: RSMEA = 0.053, SRMR = 0.045, CFI = 0.92 and TLI = 0.90. χ^2^ statistic, AIC, BIC and aBIC values were lower than those of Model 3 (Table 2). The selected model is represented in Figure 1.

The residuals of the following knowledge and attitude items showed a significant correlation (*p* < 0.001): Q9 and Q10 (r = 0.175); Q9 and Q11 (r = 0.167); Q10 and Q11 (r = 0.244); Q10 and Q13 (r = 0.311); and Q11 and Q13 (r = 0.179) (Figure 1).

The *knowledge* factor correlated positively with the factor *personal attitudes towards antibiotics* (r = 0.765; *p* < 0.001), but it showed a weak negative correlation with *patient attitudes towards healthcare provider* (r = −0.128; *p* = 0.001). The factors *personal attitudes towards antibiotics* and *patient attitudes towards healthcare provider* also showed a weak negative correlation (r = −0.181; *p* < 0.001) (Figure 1).

### 3.5. Questionnaire Internal Reliability

The questionnaire showed an acceptable internal reliability with a Cronbach alpha value of 0.62.

### 3.6. Questionnaire Acceptability

Out of 1460 distributed questionnaires, 1400 were completely or almost completely answered. Thirty-nine questionnaires were returned without answering any question and in the remaining 21 only some demographic questions were completed. Based on these figures, the calculated response rate of the Arabic/French questionnaire in Lebanon was 95.89%.

The item response rate ranged between 94% and 98% indicating a high acceptability of the questions by the Lebanese adult population.

## 4. Discussion

Knowledge, attitudes and practice (KAP) questionnaires are vital tools to assess the need, and design and evaluate intervention programs to improve the safe use of antibiotics. We reported the validation of the first Arabic/French KAP questionnaire on antibiotic use by the adult population. The questionnaire showed high content and face validity, generated reproducible results, revealed acceptable internal reliability, and was highly accepted by the Lebanese adult population.

Antibiotic resistance has knocked every part of the world with projected gloomy figures on fatality and financial losses if no appropriate and rapid action is taken worldwide [8]. Therefore, the availability of a cross-culturally validated tool that can fit populations of different languages, socioeconomic, cultural, and public health systems such as the case of Spain and Lebanon is of outmost importance as it (1) serves as a research instrument for the countries with frequently spoken languages worldwide (Arabic, French and Spanish), and (2) allows to adapt and compare intervention programs to and between different settings.

The test–retest analysis established the reproducibility of the Arabic/French questionnaire to be administered in Arabic and/or French speaking populations such as Lebanon and countries with similar cultural background. Furthermore, CFA analysis confirmed the construct validity of the questionnaire. The structure of the final model coincided with that previously validated in Spain in the identification of three factors *knowledge, personal attitudes towards antibiotics*, and *patient attitudes towards healthcare provider,* though the two models in Lebanon and Spain slightly differed in the item distribution across their respective factors. Data collected from the Lebanese adults suggested the inclusion of Q17 *“**When you buy antibiotics, the pharmacist tells you about the importance of correct therapeutic compliance/adherence”* in the *knowledge* section as well as in the *patient attitudes towards healthcare provider* factor. On the one hand, this could be explained in part by the limited knowledge of the Lebanese patients and pharmacists regarding antibiotics and on the other hand by the reliance of the Lebanese patients on pharmacists as a source of medicines in case of illness. In Lebanon, a considerable proportion of pharmacists believes that antibiotics are not harmful and may deliver these drugs to customers without referring them to a physician, in order to retain these customers and increase economic profit [32]. On their side, Lebanese people accept willingly an antibiotic recommended by a pharmacist [32]. Access to healthcare is limited in Lebanon with a health system mainly based on private healthcare and paid by out-of-pocket money [33]. Individuals with lower socioeconomic status use pharmacists as an outlet of medicines to avoid expenses of medical consultation and clinical tests, even though most Lebanese people consider that the personnel working as drug dispenser is not sufficiently qualified [34]. A recent study in Lebanon showed that over three-quarters of the participants consider their pharmacists responsible for the safety and security of their medication [34]. Therefore, individuals with a limited knowledge on antibiotics might buy these drugs without prescription [35], and not be adequately advised about the use of antibiotics by pharmacists. Pharmacists in Lebanon are not used to advise patients on the correct use of medicine. Indeed, less than 20% of the pharmacists in the country consider themselves as patient counsellors [36].

The model chosen for the Lebanese population encompassed a correlation between the residuals of certain items. This finding was expected as item cross-loading and correlation of item residuals is a feature of questionnaires similar to ours that measures more than one factor, i.e., *knowledge, personal attitudes towards antibiotics* and *attitudes towards healthcare provider* factors [37].

Our study has an inherent limitation that lies in the absence of a gold standard, i.e., an instrument superior to ours, such as a measurement of a biologic factor, to which the performance could be compared. Hence the concurrent validity, i.e., how well our questionnaire compares to a well-established instrument, as is often the case in epidemiologic studies, cannot be tested here.

## 5. Conclusions

The established validity of the original Spanish questionnaire in a socioeconomically different country (Lebanon), and in different languages (Arabic and French) demonstrated its robustness to variations of settings and showed its feasibility and acceptability. Our multilingual tool could prove useful in establishing education programs in countries of different socio-economic contexts.

## Figures and Tables

**Figure 1 ijerph-19-00687-f001:**
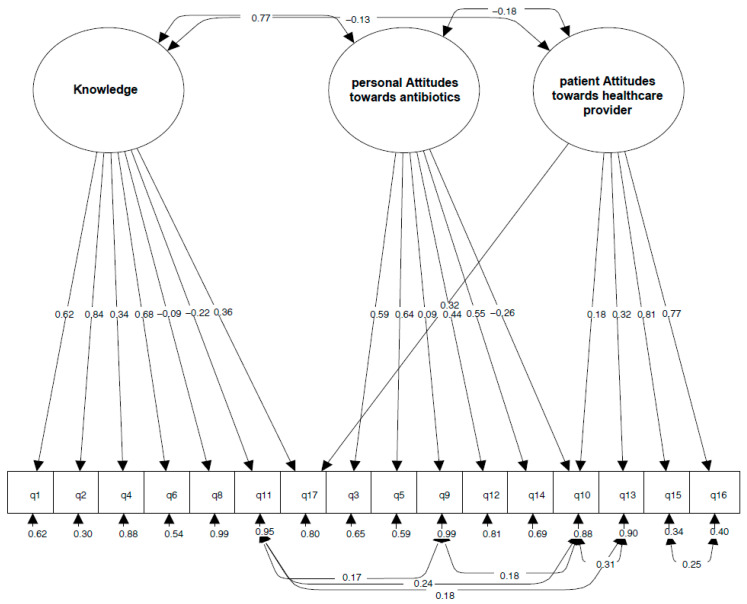
Representation of the knowledge and attitudes model selected by CFA analysis (Model 4). The Figure represents each of the three factors (*knowledge, personal attitudes towards antibiotics* and *patient attitudes towards healthcare provider*) with its respective standardized item loadings and item residuals. Double-sided arrows represent correlations between the variables. Single headed arrows represent the correlation of each item with its corresponding factor(s).

**Table 1 ijerph-19-00687-t001:** Test–retest reliability assessment of knowledge and attitudes statements of the KAP questionnaire on antibiotic use.

Knowledge or Attitude Statement		ICC (95%CI)
Q1	Antibiotics are effective against viruses	0.80 (0.69, 0.87)
Q2	When I get a cold, I take antibiotics to help me feel better faster	0.77 (0.66, 0.85)
Q3	If I feel better after a few days, I sometimes stop taking my antibiotics before completing the course of treatment	0.82 (0.73, 0.88)
Q4	I expect my doctor to prescribe antibiotics if I suffer from common cold or flu symptoms	0.71 (0.56, 0.81)
Q5	It is good to be able to get antibiotics from relatives or friends without having to see a medical doctor	0.82 (0.73, 0.88)
Q6	When I have a sore throat, I prefer to use an antibiotic	0.87 (0.81, 0.92)
Q7	Each infection needs a different antibiotic	0.76 (0.64, 0.84)
Q8	Antibiotics can kill the bacteria that normally live on the skin and in the gut	0.72 (0.57, 0.82)
Q9	If I feel side effects during a course of treatment of antibiotics, I should stop taking them as soon as possible	0.85 (0.77, 0.90)
Q10	I take the antibiotics according to the doctor’s instructions	0.71 (0.55, 0.81)
Q11	If antibiotics are consumed in excess, they will not work when they are really needed	0.70 (0.55, 0.80)
Q12	I prefer to keep antibiotics at home in case there is a need for them later	0.79 (0.69, 0.86)
Q13	I trust the doctor’s decision if s/he decides to prescribe or not prescribe antibiotics	0.59 (0.38, 0.73)
Q14	If I believe that I need an antibiotic and the doctor did not prescribe it, I will get it at the pharmacy without a prescription	0.71 (0.56, 0.81)
Q15	Doctors often explain clearly to the patient the reasons for prescribing or not prescribing antibiotics	0.78 (0.66, 0.85)
Q16	Doctors often explain clearly to the patient the instructions for the use of antibiotics	0.80 (0.70, 0.87)
Q17	When you buy antibiotics, the pharmacist tells you about the importance of correct therapeutic compliance/adherence	0.87 (0.80, 0.91)

ICC: intraclass correlation coefficient; CI: confidence interval.

**Table 2 ijerph-19-00687-t002:** Comparison of the goodness of fit parameters between the tested models of the knowledge and attitude items of the KAP questionnaire on antibiotic use.

Indicator	Model 1	Model 2	Model 3	Model 4
χ^2^	1185.56	852.603	629.856	459.392
df	116	100	97	93
*p*	<0.001	<0.001	<0.001	<0.001
RSMEA (90% CI)	0.081 (0.077, 0.085)	0.073 (0.069, 0.078)	0.063 (0.058, 0.067)	0.053 (0.048, 0.058)
CFI	0.77	0.83	0.88	0.92
TLI	0.73	0.80	0.85	0.90
AIC	112,294.425	105,927.216	105,710.469	105,548.005
BIC	112,577.613	106,199.916	105,998.901	105,857.414
aBIC	112,406.075	106,034.731	105,824.187	105,669.993
SRMR	0.085	0.068	0.055	0.045

χ^2^: Chi-square value; df: degrees of freedom; *p*: *p*-value (Chi-square); RSMEA: root mean squared error approximation; CFI: comparative fit index; TLI: Tucker–Lewis index; AIC: Akaike information criterion, BIC: Bayesian information criterion; aBIC: sample-size adjusted BIC; SRMR: standardized root mean square residual.

**Table 3 ijerph-19-00687-t003:** Factor loadings and standard errors from the three-factor model of the knowledge and attitude items of the KAP questionnaire on antibiotic use.

Item	Loading Estimate	Standard Error	*p*-Value	Standard Loading Estimate
Knowledge				
Q1. Antibiotics are effective against viruses	1.078	0.036	<0.001	0.618
Q2. When I get a cold, I take antibiotics to help me feel better faster	1.212	0.032	<0.001	0.839
Q4. I expect my doctor to prescribe antibiotics if I suffer from common cold or flu symptoms	0.666	0.047	<0.001	0.342
Q6. When I have a sore throat, I prefer to use an antibiotic	1.044	0.034	<0.001	0.682
Q8. Antibiotics can kill the bacteria that normally live on the skin and in the gut	−0.121	0.042	0.004	−0.086
Q11. If antibiotics are consumed in excess, they will not work when they are really needed	−0.257	0.034	<0.001	−0.222
Q17. When you buy antibiotics, the pharmacist tells you about the importance of correct therapeutic compliance/adherence	0.545	0.046	<0.001	0.363
Personal attitudes towards antibiotics				
Q3. If I feel better after a few days, I sometimes stop taking my antibiotics before completing the course of treatment	1.453	0.063	<0.001	0.590
Q5. It is good to be able to get antibiotics from relatives or friends without having to see a medical doctor	1.037	0.045	<0.001	0.640
Q9. If I feel side effects during a course of treatment of antibiotics, I should stop taking them as soon as possible	0.208	0.073	0.004	0.087
Q10. I take the antibiotics according to the doctor’s instructions	−0.315	0.037	<0.001	−0.263
Q12. I prefer to keep antibiotics at home in case there is a need for them later	1.186	0.067	<0.001	0.438
Q14. If I believe that I need an antibiotic and the doctor did not prescribe it, I will get it at the pharmacy without a prescription	1.116	0.052	<0.001	0.553
Patient attitudes towards healthcare provider				
Q10. I take the antibiotics according to the doctor’s instructions	0.207	0.046	<0.001	0.175
Q13. I trust the doctor’s decision if s/he decides to prescribe or not prescribe antibiotics	0.566	0.091	<0.001	0.318
Q15. Doctors often explain clearly to the patient the reasons for prescribing or not prescribing antibiotics	1.791	0.124	<0.001	0.813
Q16. Doctors often explain clearly to the patient the instructions for the use of antibiotics	1.686	0.120	<0.001	0.772
Q17. When you buy antibiotics, the pharmacist tells you about the importance of correct therapeutic compliance/adherence	0.751	0.117	<0.001	0.319

## Data Availability

The data that support the findings of this study are available on request from the corresponding author.

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
