# Peer review of "Development and Validation of a Knowledge, Attitude and Practice Questionnaire on Antibiotic Use in Arabic and French Languages in Lebanon"

_ijerph, 2022, doi:10.3390/ijerph19020687_

Round 1

Reviewer 1 Report

The topic of the article is not new. The article touches on the problem of awareness of the abuse of antibiotics, known for many years, but still very important for clinicians.

In their article, the authors attempted to verify the suitability in Lebanon of the KAP questionnaire used in Spain, necessary for the design and evaluation of intervention programs for the use of antibiotics.

The assumed goal was achieved, although the authors used only people living in a big city for their research. The authors proved that the KAP questionnaire can be easily and quickly adapted to the needs of the country in which it will be used.

Reviewer 2 Report

The aim of the article was to validate a previously developed KAP-questionnaire on antibiotic misuse into Arabic and French. The translated questionnaire was validated in 2019 in a population of adults in Lebanon.
The modified kappa (K*) statistic was higher of 0.75 and equal to I-CVI, showing an agreement on the clarity and the relevancy of the items of the questionnaire. Also, the S-CVI/Ave was greater than or equal to 0.95, confirming the content validity of the scale..
To conclude, the authors demonstrated the questionnaire robustness to variations of settings and showed its feasibility and acceptability.

Minor revisions:

Please, provide Figure 1 of higher quality.

Reviewer 3 Report

The paper is very well structured and written. It is an essential contribution to advancing the study of antibiotic resistance and the vast and globally significant Arabic-speaking community.

As the paper is primarily methodological, it is worth refining all the methodological elements in the article.

I suggest that the authors of the paper pay attention to the following in the Methods chapter:

  1. I suggest moving or repeating the sentence, information from subsection 2.8, starting with "The questionnaire was..." in subsection 2.1, where the study population is mentioned.
  2. Additionally, in section 2.1, in the second paragraph, it should be explained what percentage of the population of adult parents who have children in school were the 1460 people surveyed.
  3. Clarify the method used to deliver the questionnaires to parents. One might guess that they were paper questionnaires, but was that the case? Did parents fill out these questionnaires at school, or did they take them home? Who distributed the questionnaires, and who collected them back?
  4. need to clarify why these 11 schools were selected: was it purposeful or random? How were these schools different? What were the proportions of questionnaires collected from these schools?
  5. section 2.1 also needs to write what number of questionnaires were sent to parents and returned? Did each school get the same number of questionnaires or not?
